# Distributionally Robust Logistic Regression

**Soroosh Shafieezadeh-Abadeh**     **Peyman Mohajerin Esfahani**     **Daniel Kuhn**
École Polytechnique Fédérale de Lausanne, CH-1015 Lausanne, Switzerland
{soroosh.shafiee,peyman.mohajerin,daniel.kuhn} @epfl.ch

## Abstract

This paper proposes a distributionally robust approach to logistic regression. We use the Wasserstein distance to construct a ball in the space of probability distributions centered at the uniform distribution on the training samples. If the radius of this ball is chosen judiciously, we can guarantee that it contains the unknown data-generating distribution with high confidence. We then formulate a distributionally robust logistic regression model that minimizes a worst-case expected logloss function, where the worst case is taken over all distributions in the Wasserstein ball. We prove that this optimization problem admits a tractable reformulation and encapsulates the classical as well as the popular regularized logistic regression problems as special cases. We further propose a distributionally robust approach based on Wasserstein balls to compute upper and lower confidence bounds on the misclassification probability of the resulting classifier. These bounds are given by the optimal values of two highly tractable linear programs. We validate our theoretical out-of-sample guarantees through simulated and empirical experiments.

## 1 Introduction

Logistic regression is one of the most frequently used classification methods [1]. Its objective is to establish a probabilistic relationship between a continuous feature vector and a binary explanatory variable. However, in spite of its overwhelming success in machine learning, data analytics and medicine etc., logistic regression models can display a poor out-of-sample performance if training data is sparse. In this case modelers often resort to *ad hoc* regularization techniques in order to combat overfitting effects. This paper aims to develop new regularization techniques for logistic regression—and to provide intuitive probabilistic interpretations for existing ones—by using tools from modern distributionally robust optimization.

**Logistic Regression:**   Let $x \in \mathbb{R}^n$ denote a feature vector and $y \in \{-1, +1\}$ the associated binary label to be predicted. In logistic regression, the conditional distribution of $y$ given $x$ is modeled as

$$\mathrm{Prob}(y|x) = [1 + \exp(-y\langle \beta, x \rangle)]^{-1} \,, \tag{1}$$

where the weight vector $\beta \in \mathbb{R}^n$ constitutes an unknown regression parameter. Suppose that $N$ training samples $\{(\hat{x}_i, \hat{y}_i)\}_{i=1}^N$ have been observed. Then, the maximum likelihood estimator of classical logistic regression is found by solving the geometric program

$$\min_{\beta} \frac{1}{N} \sum_{i=1}^N l_\beta(\hat{x}_i, \hat{y}_i) \,, \tag{2}$$

whose objective function is given by the sample average of the *logloss function* $l_\beta(x, y) = \log(1 + \exp(-y\langle \beta, x \rangle))$. It has been observed, however, that the resulting maximum likelihood estimator may display a poor out-of-sample performance. Indeed, it is well documented that minimizing the average logloss function leads to overfitting and weak classification performance [2, 3]. In order

to overcome this deficiency, it has been proposed to modify the objective function of problem (2) [4, 5, 6]. An alternative approach is to add a regularization term to the logloss function in order to mitigate overfitting. These regularization techniques lead to a modified optimization problem

$$\min_{\beta} \frac{1}{N} \sum_{i=1}^{N} l_{\beta}(\hat{x}_i, \hat{y}_i) + \varepsilon R(\beta)\,, \tag{3}$$

where $R(\beta)$ and $\varepsilon$ denote the regularization function and the associated coefficient, respectively. A popular choice for the regularization term is $R(\beta) = \|\beta\|$, where $\|\cdot\|$ denotes a generic norm such as the $\ell_1$ or the $\ell_2$-norm. The use of $\ell_1$-regularization tends to induce sparsity in $\beta$, which in turn helps to combat overfitting effects [7]. Moreover, $\ell_1$-regularized logistic regression serves as an effective means for feature selection. It is further shown in [8] that $\ell_1$-regularization outperforms $\ell_2$-regularization when the number of training samples is smaller than the number of features. On the downside, $\ell_1$-regularization leads to non-smooth optimization problems, which are more challenging. Algorithms for large scale regularized logistic regression are discussed in [9, 10, 11, 12].

**Distributionally Robust Optimization:**  Regression and classification problems are typically modeled as optimization problems under uncertainty. To date, optimization under uncertainty has been addressed by several complementary modeling paradigms that differ mainly in the representation of uncertainty. For instance, stochastic programming assumes that the uncertainty is governed by a known probability distribution and aims to minimize a probability functional such as the expected cost or a quantile of the cost distribution [13, 14]. In contrast, robust optimization ignores all distributional information and aims to minimize the worst-case cost under all possible uncertainty realizations [15, 16, 17]. While stochastic programs may rely on distributional information that is not available or hard to acquire in practice, robust optimization models may adopt an overly pessimistic view of the uncertainty and thereby promote over-conservative decisions.

The emerging field of distributionally robust optimization aims to bridge the gap between the conservatism of robust optimization and the specificity of stochastic programming: it seeks to minimize a worst-case probability functional (e.g., the worst-case expectation), where the worst case is taken with respect to an ambiguity set, that is, a family of distributions consistent with the given prior information on the uncertainty. The vast majority of the existing literature focuses on ambiguity sets characterized through moment and support information, see e.g. [18, 19, 20]. However, ambiguity sets can also be constructed via distance measures in the space of probability distributions such as the Prohorov metric [21] or the Kullback-Leibler divergence [22]. Due to its attractive measure concentration properties, we use here the Wasserstein metric to construct ambiguity sets.

**Contribution:**  In this paper we propose a distributionally robust perspective on logistic regression. Our research is motivated by the well-known observation that regularization techniques can improve the out-of-sample performance of many classifiers. In the context of support vector machines and Lasso, there have been several recent attempts to give *ad hoc* regularization techniques a robustness interpretation [23, 24]. However, to the best of our knowledge, no such connection has been established for logistic regression. In this paper we aim to close this gap by adopting a new distributionally robust optimization paradigm based on Wasserstein ambiguity sets [25]. Starting from a data-driven distributionally robust statistical learning setup, we will derive a family of regularized logistic regression models that admit an intuitive probabilistic interpretation and encapsulate the classical regularized logistic regression (3) as a special case. Moreover, by invoking recent measure concentration results, our proposed approach provides a probabilistic guarantee for the emerging regularized classifiers, which seems to be the first result of this type. All proofs are relegated to the technical appendix. We summarize our main contributions as follows:

- **Distributionally robust logistic regression model and tractable reformulation:** We propose a data-driven distributionally robust logistic regression model based on an ambiguity set induced by the Wasserstein distance. We prove that the resulting semi-infinite optimization problem admits an equivalent reformulation as a tractable convex program.
- **Risk estimation:** Using similar distributionally robust optimization techniques based on the Wasserstein ambiguity set, we develop two highly tractable linear programs whose optimal values provide confidence bounds on the misclassification probability or *risk* of the emerging classifiers.
- **Out-of-sample performance guarantees:** Adopting a distributionally robust framework allows us to invoke results from the measure concentration literature to derive finite-sample probabilistic

guarantees. Specifically, we establish *out-of-sample* performance guarantees for the classifiers obtained from the proposed distributionally robust optimization model.

- **Probabilistic interpretation of existing regularization techniques:** We show that the standard regularized logistic regression is a special case of our framework. In particular, we show that the regularization coefficient $\varepsilon$ in (3) can be interpreted as the size of the ambiguity set underlying our distributionally robust optimization model.

## 2 A distributionally robust perspective on statistical learning

In the standard statistical learning setting all training and test samples are drawn independently from some distribution $\mathbb{P}$ supported on $\Xi = \mathbb{R}^n \times \{-1, +1\}$. If the distribution $\mathbb{P}$ was known, the best weight parameter $\beta$ could be found by solving the stochastic optimization problem

$$\inf_{\beta} \left\{ \mathbb{E}^{\mathbb{P}} \left[ l_\beta(x, y) \right] = \int_{\mathbb{R}^n \times \{-1,+1\}} l_\beta(x, y) \mathbb{P}(\mathrm{d}(x, y)) \right\}. \tag{4}$$

In practice, however, $\mathbb{P}$ is only indirectly observable through $N$ independent training samples. Thus, the distribution $\mathbb{P}$ is itself uncertain, which motivates us to address problem (4) from a distributionally robust perspective. This means that we use the training samples to construct an ambiguity set $\mathcal{P}$, that is, a family of distributions that contains the unknown distribution $\mathbb{P}$ with high confidence. Then we solve the distributionally robust optimization problem

$$\inf_{\beta} \sup_{\mathbb{Q} \in \mathcal{P}} \mathbb{E}^{\mathbb{Q}} \left[ l_\beta(x, y) \right], \tag{5}$$

which minimizes the worst-case expected logloss function. The construction of the ambiguity set $\mathcal{P}$ should be guided by the following principles. *(i) Tractability:* It must be possible to solve the distributionally robust optimization problem (5) efficiently. *(ii) Reliability:* The optimizer of (5) should be near-optimal in (4), thus facilitating attractive out-of-sample guarantees. *(iii) Asymptotic consistency:* For large training data sets, the solution of (5) should converge to the one of (4). In this paper we propose to use the Wasserstein metric to construct $\mathcal{P}$ as a ball in the space of probability distributions that satisfies *(i)–(iii)*.

**Definition 1** (Wasserstein Distance)**.** Let $M(\Xi^2)$ denote the set of probability distributions on $\Xi \times \Xi$. The Wasserstein distance between two distributions $\mathbb{P}$ and $\mathbb{Q}$ supported on $\Xi$ is defined as

$$W(\mathbb{Q}, \mathbb{P}) := \inf_{\Pi \in M(\Xi^2)} \left\{ \int_{\Xi^2} d(\xi, \xi') \, \Pi(\mathrm{d}\xi, \mathrm{d}\xi') \ : \ \Pi(\mathrm{d}\xi, \Xi) = \mathbb{Q}(\mathrm{d}\xi), \ \Pi(\Xi, \mathrm{d}\xi') = \mathbb{P}(\mathrm{d}\xi') \right\},$$

where $\xi = (x, y)$ and $d(\xi, \xi')$ is a metric on $\Xi$.

The Wasserstein distance represents the minimum cost of moving the distribution $\mathbb{P}$ to the distribution $\mathbb{Q}$, where the cost of moving a unit mass from $\xi$ to $\xi'$ amounts to $d(\xi, \xi')$.

In the remainder, we denote by $\mathbb{B}_\varepsilon(\mathbb{P}) := \{\mathbb{Q} : W(\mathbb{Q}, \mathbb{P}) \le \varepsilon\}$ the ball of radius $\varepsilon$ centered at $\mathbb{P}$ with respect to the Wasserstein distance. In this paper we propose to use Wasserstein balls as ambiguity sets. Given the training data points $\{(\hat{x}_i, \hat{y}_i)\}_{i=1}^N$, a natural candidate for the center of the Wasserstein ball is the empirical distribution $\hat{\mathbb{P}}_N = \frac{1}{N} \sum_{i=1}^N \delta_{(\hat{x}_i, \hat{y}_i)}$, where $\delta_{(\hat{x}_i, \hat{y}_i)}$ denotes the Dirac point measure at $(\hat{x}_i, \hat{y}_i)$. Thus, we henceforth examine the distributionally robust optimization problem

$$\inf_{\beta} \sup_{\mathbb{Q} \in \mathbb{B}_\varepsilon(\hat{\mathbb{P}}_N)} \mathbb{E}^{\mathbb{Q}} \left[ l_\beta(x, y) \right] \tag{6}$$

equipped with a Wasserstein ambiguity set. Note that (6) reduces to the average logloss minimization problem (2) associated with classical logistic regression if we set $\varepsilon = 0$.

## 3 Tractable reformulation and probabilistic guarantees

In this section we demonstrate that (6) can be reformulated as a tractable convex program and establish probabilistic guarantees for its optimal solutions.

## 3.1 Tractable reformulation

We first define a metric on the feature-label space, which will be used in the remainder.

**Definition 2** (Metric on the Feature-Label Space). The distance between two data points $(x, y), (x', y') \in \Xi$ is defined as $d\big((x, y), (x', y')\big) = \|x - x'\| + \kappa |y - y'|/2$, where $\|\cdot\|$ is any norm on $\mathbb{R}^n$, and $\kappa$ is a positive weight.

The parameter $\kappa$ in Definition 2 represents the relative emphasis between feature mismatch and label uncertainty. The following theorem presents a tractable reformulation of the distributionally robust optimization problem (6) and thus constitutes the first main result of this paper.

**Theorem 1** (Tractable Reformulation). The optimization problem (6) is equivalent to

$$\hat{J} := \inf_{\beta} \sup_{\mathbb{Q} \in \mathbb{B}_\varepsilon(\hat{\mathbb{P}}_N)} \mathbb{E}^{\mathbb{Q}}\left[l_\beta(x, y)\right] = \begin{cases} \min_{\beta, \lambda, s_i} & \lambda \varepsilon + \frac{1}{N} \sum_{i=1}^{N} s_i \\ \text{s.t.} & l_\beta(\hat{x}_i, \hat{y}_i) \leq s_i & \forall i \leq N \\ & l_\beta(\hat{x}_i, -\hat{y}_i) - \lambda \kappa \leq s_i & \forall i \leq N \\ & \|\beta\|_* \leq \lambda. \end{cases} \tag{7}$$

Note that (7) constitutes a tractable convex program for most commonly used norms $\|\cdot\|$.

**Remark 1** (Regularized Logistic Regression). As the parameter $\kappa > 0$ characterizing the metric $d(\cdot, \cdot)$ tends to infinity, the second constraint group in the convex program (7) becomes redundant. Hence, (7) reduces to the celebrated regularized logistic regression problem

$$\inf_{\beta} \varepsilon \|\beta\|_* + \frac{1}{N} \sum_{i=1}^{N} l_\beta(\hat{x}_i, \hat{y}_i),$$

where the regularization function is determined by the dual norm on the feature space, while the regularization coefficient coincides with the radius of the Wasserstein ball. Note that for $\kappa = \infty$ the Wasserstein distance between two distributions is infinite if they assign different labels to a fixed feature vector with positive probability. Any distribution in $\mathbb{B}_\varepsilon(\hat{\mathbb{P}}_N)$ must then have non-overlapping conditional supports for $y = +1$ and $y = -1$. Thus, setting $\kappa = \infty$ reflects the belief that the label is a (deterministic) function of the feature and that label measurements are exact. As this belief is not tenable in most applications, an approach with $\kappa < \infty$ may be more satisfying.

## 3.2 Out-of-sample performance guarantees

We now exploit a recent measure concentration result characterizing the speed at which $\hat{\mathbb{P}}_N$ converges to $\mathbb{P}$ with respect to the Wasserstein distance [26] in order to derive out-of-sample performance guarantees for distributionally robust logistic regression.

In the following, we let $\hat{\Xi}_N := \{(\hat{x}_i, \hat{y}_i)\}_{i=1}^{N}$ be a set of $N$ independent training samples from $\mathbb{P}$, and we denote by $\hat{\beta}, \hat{\lambda}$, and $\hat{s}_i$ the optimal solutions and $\hat{J}$ the corresponding optimal value of (7). Note that these values are random objects as they depend on the random training data $\hat{\Xi}_N$.

**Theorem 2** (Out-of-Sample Performance). Assume that the distribution $\mathbb{P}$ is light-tailed, i.e., there is $a > 1$ with $A := \mathbb{E}^{\mathbb{P}}[\exp(\|2x\|^a)] < +\infty$. If the radius $\varepsilon$ of the Wasserstein ball is set to

$$\varepsilon_N(\eta) = \left(\frac{\log\left(c_1 \eta^{-1}\right)}{c_2 N}\right)^{\frac{1}{a}} \mathbb{1}_{\left\{N < \frac{\log\left(c_1 \eta^{-1}\right)}{c_2 c_3}\right\}} + \left(\frac{\log\left(c_1 \eta^{-1}\right)}{c_2 N}\right)^{\frac{1}{n}} \mathbb{1}_{\left\{N \geq \frac{\log\left(c_1 \eta^{-1}\right)}{c_2 c_3}\right\}}, \tag{8}$$

then we have $\mathbb{P}^N\left\{\mathbb{P} \in \mathbb{B}_\varepsilon(\hat{\mathbb{P}}_N)\right\} \geq 1 - \eta$, implying that $\mathbb{P}^N\{\hat{\Xi}_N : \mathbb{E}^{\mathbb{P}}[l_{\hat{\beta}}(x, y)] \leq \hat{J}\} \geq 1 - \eta$ for all sample sizes $N \geq 1$ and confidence levels $\eta \in (0, 1]$. Moreover, the positive constants $c_1, c_2$, and $c_3$ appearing in (8) depend only on the light-tail parameters $a$ and $A$, the dimension $n$ of the feature space, and the metric on the feature-label space.

**Remark 2** (Worst-Case Loss). Denoting the empirical logloss function on the training set $\hat{\Xi}_N$ by $\mathbb{E}^{\hat{\mathbb{P}}^N}[l_{\hat{\beta}}(x, y)]$, the worst-case loss $\hat{J}$ can be expressed as

$$\hat{J} = \hat{\lambda} \varepsilon + \mathbb{E}^{\hat{\mathbb{P}}^N}[l_{\hat{\beta}}(x, y)] + \frac{1}{N} \sum_{i=1}^{N} \max\{0, \hat{y}_i \langle \hat{\beta}, \hat{x}_i \rangle - \hat{\lambda} \kappa\}. \tag{9}$$

Note that the last term in (9) can be viewed as a complementary regularization term that does not appear in standard regularized logistic regression. This term accounts for label uncertainty and decreases with $\kappa$. Thus, $\kappa$ can be interpreted as our trust in the labels of the training samples. Note that this regularization term vanishes for $\kappa \to \infty$. One can further prove that $\hat{\lambda}$ converges to $\|\hat{\beta}\|_*$ for $\kappa \to \infty$, implying that (9) reduces to the standard regularized logistic regression in this limit.

**Remark 3** (Performance Guarantees). The following comments are in order:

I. **Light-Tail Assumption:** The light-tail assumption of Theorem 2 is restrictive but seems to be unavoidable for any a priori guarantees of the type described in Theorem 2. Note that this assumption is automatically satisfied if the features have bounded support or if they are known to follow, for instance, a Gaussian or exponential distribution.

II. **Asymptotic Consistency:** For any fixed confidence level $\eta$, the radius $\varepsilon_N(\eta)$ defined in (8) drops to zero as the sample size $N$ increases, and thus the ambiguity set shrinks to a singleton. To be more precise, with probability 1 across all training datasets, a sequence of distributions in the ambiguity set (8) converges in the Wasserstein metric, and thus weakly, to the unknown data generating distribution $\mathbb{P}$; see [25, Corollary 3.4] for a formal proof. Consequently, the solution of (2) can be shown to converge to the solution of (4) as $N$ increases.

III. **Finite Sample Behavior:** The a priori bound (8) on the size of the Wasserstein ball has two growth regimes. For small $N$, the radius decreases as $N^{\frac{1}{a}}$, and for large $N$ it scales with $N^{\frac{1}{n}}$, where $n$ is the dimension of the feature space. We refer to [26, Section 1.3] for further details on the optimality of these rates and potential improvements for special cases. Note that when the support of the underlying distribution $\mathbb{P}$ is bounded or $\mathbb{P}$ has a Gaussian distribution, the parameter $a$ can be effectively set to 1.

### 3.3 Risk Estimation: Worst- and Best-Cases

One of the main objectives in logistic regression is to control the classification performance. Specifically, we are interested in *predicting* labels from features. This can be achieved via a classifier function $f_\beta : \mathbb{R}^n \to \{+1, -1\}$, whose *risk* $\mathfrak{R}(\beta) := \mathbb{P}[y \neq f_\beta(x)]$ represents the misclassification probability. In logistic regression, a natural choice for the classifier is $f_\beta(x) = +1$ if $\mathrm{Prob}(+1|x) > 0.5; = -1$ otherwise. The conditional probability $\mathrm{Prob}(y|x)$ is defined in (1). The risk associated with this classifier can be expressed as $\mathfrak{R}(\beta) = \mathbb{E}^{\mathbb{P}}[\mathbb{1}_{\{y\langle\beta,x\rangle \leq 0\}}]$. As in Section 3.1, we can use worst- and best-case expectations over Wasserstein balls to construct confidence bounds on the risk.

**Theorem 3** (Risk Estimation). For any $\hat{\beta}$ depending on the training dataset $\{(\hat{x}_i, \hat{y}_i)\}_{i=1}^N$ we have:

(i) The worst-case risk $\mathfrak{R}_{\max}(\hat{\beta}) := \sup_{\mathbb{Q}\in\mathbb{B}_\varepsilon(\hat{\mathbb{P}}_N)} \mathbb{E}^{\mathbb{Q}}[\mathbb{1}_{\{y\langle\hat{\beta},x\rangle\leq 0\}}]$ is given by

$$
\mathfrak{R}_{\max}(\hat{\beta}) = \begin{cases} \min_{\lambda,s_i,r_i,t_i} & \lambda\varepsilon + \frac{1}{N}\sum_{i=1}^N s_i \\ \text{s.t.} & 1 - r_i\hat{y}_i\langle\hat{\beta},\hat{x}_i\rangle \leq s_i & \forall i \leq N \\ & 1 + t_i\hat{y}_i\langle\hat{\beta},\hat{x}_i\rangle - \lambda\kappa \leq s_i & \forall i \leq N \\ & r_i\|\hat{\beta}\|_* \leq \lambda, \quad t_i\|\hat{\beta}\|_* \leq \lambda & \forall i \leq N \\ & r_i,t_i,s_i \geq 0 & \forall i \leq N. \end{cases} \tag{10a}
$$

If the Wasserstein radius $\varepsilon$ is set to $\varepsilon_N(\eta)$ as defined in (8), then $\mathfrak{R}_{\max}(\hat{\beta}) \geq \mathfrak{R}(\hat{\beta})$ with probability $1 - \eta$ across all training sets $\{(x_i, y_i)\}_{i=1}^N$.

(ii) Similarly, the best-case risk $\mathfrak{R}_{\min}(\hat{\beta}) := \inf_{\mathbb{Q}\in\mathbb{B}_\varepsilon(\hat{\mathbb{P}}_N)} \mathbb{E}^{\mathbb{Q}}[\mathbb{1}_{\{y\langle\hat{\beta},x\rangle<0\}}]$ is given by

$$
\mathfrak{R}_{\min}(\hat{\beta}) = 1 - \begin{cases} \min_{\lambda,s_i,r_i,t_i} & \lambda\varepsilon + \frac{1}{N}\sum_{i=1}^N s_i \\ \text{s.t.} & 1 + r_i\hat{y}_i\langle\hat{\beta},\hat{x}_i\rangle \leq s_i & \forall i \leq N \\ & 1 - t_i\hat{y}_i\langle\hat{\beta},\hat{x}_i\rangle - \lambda\kappa \leq s_i & \forall i \leq N \\ & r_i\|\hat{\beta}\|_* \leq \lambda, \quad t_i\|\hat{\beta}\|_* \leq \lambda & \forall i \leq N \\ & r_i,t_i,s_i \geq 0 & \forall i \leq N. \end{cases} \tag{10b}
$$

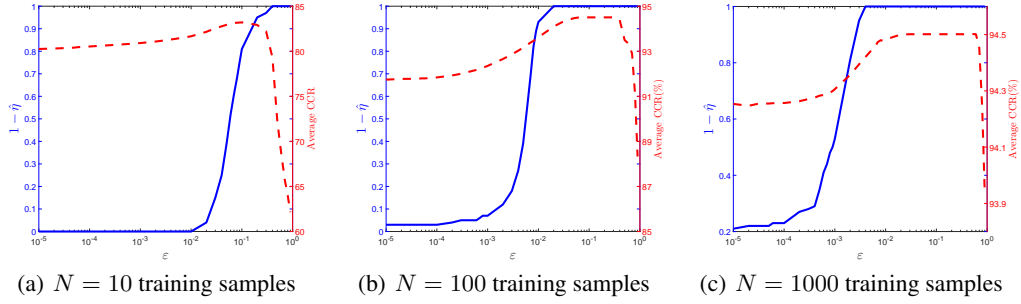

| (a) $N = 10$ training samples | (b) $N = 100$ training samples | (c) $N = 1000$ training samples |

Figure 1: Out-of-sample performance (solid blue line) and the average CCR (dashed red line)

If the Wasserstein radius $\varepsilon$ is set to $\varepsilon_N(\eta)$ as defined in (8), then $\mathfrak{R}_{\min}(\hat{\beta}) \leq \mathfrak{R}(\hat{\beta})$ with probability $1 - \eta$ across all training sets $\{(x_i, y_i)\}_{i=1}^N$.

We emphasize that (10a) and (10b) constitute highly tractable linear programs. Moreover, we have $\mathfrak{R}_{\min}(\hat{\beta}) \leq \mathfrak{R}(\hat{\beta}) \leq \mathfrak{R}_{\max}(\hat{\beta})$ with probability $1 - 2\eta$.

## 4 Numerical Results

We now showcase the power of distributionally robust logistic regression in simulated and empirical experiments. All optimization problems are implemented in MATLAB via the modeling language YALMIP [27] and solved with the state-of-the-art nonlinear programming solver IPOPT [28]. All experiments were run on an Intel XEON CPU (3.40GHz). For the largest instance studied ($N = 1000$), the problems (2), (3), (7) and (10) were solved in 2.1, 4.2, 9.2 and 0.05 seconds, respectively.

### 4.1 Experiment 1: Out-of-Sample Performance

We use a simulation experiment to study the out-of-sample performance guarantees offered by distributionally robust logistic regression. As in [8], we assume that the features $x \in \mathbb{R}^{10}$ follow a multivariate standard normal distribution and that the conditional distribution of the labels $y \in \{+1, -1\}$ is of the form (1) with $\beta = (10, 0, \ldots, 0)$. The true distribution $\mathbb{P}$ is uniquely determined by this information. If we use the $\ell_\infty$-norm to measure distances in the feature space, then $\mathbb{P}$ satisfies the light-tail assumption of Theorem 2 for $2 > a \gtrsim 1$. Finally, we set $\kappa = 1$.

Our experiment comprises 100 simulation runs. In each run we generate $N \in \{10, 10^2, 10^3\}$ training samples and $10^4$ test samples from $\mathbb{P}$. We calibrate the distributionally robust logistic regression model (6) to the training data and use the test data to evaluate the average logloss as well as the correct classification rate (CCR) of the classifier associated with $\hat{\beta}$. We then record the percentage $\hat{\eta}_N(\varepsilon)$ of simulation runs in which the average logloss exceeds $\hat{J}$. Moreover, we calculate the average CCR across all simulation runs. Figure 1 displays both $1 - \hat{\eta}_N(\varepsilon)$ and the average CCR as a function of $\varepsilon$ for different values of $N$. Note that $1 - \hat{\eta}_N(\varepsilon)$ quantifies the probability (with respect to the training data) that $\mathbb{P}$ belongs to the Wasserstein ball of radius $\varepsilon$ around the empirical distribution $\hat{\mathbb{P}}_N$. Thus, $1 - \hat{\eta}_N(\varepsilon)$ increases with $\varepsilon$. The average CCR benefits from the regularization induced by the distributional robustness and increases with $\varepsilon$ as long as the empirical confidence $1 - \hat{\eta}_N(\varepsilon)$ is smaller than 1. As soon as the Wasserstein ball is large enough to contain the distribution $\mathbb{P}$ with high confidence ($1 - \hat{\eta}_N(\varepsilon) \lesssim 1$), however, any further increase of $\varepsilon$ is detrimental to the average CCR.

Figure 1 also indicates that the radius $\varepsilon$ implied by a fixed empirical confidence level scales inversely with the number of training samples $N$. Specifically, for $N = 10, 10^2, 10^3$, the Wasserstein radius implied by the confidence level $1 - \hat{\eta} = 95\%$ is given by $\varepsilon \approx 0.2, 0.02, 0.003$, respectively. This observation is consistent with the a priori estimate (8) of the Wasserstein radius $\varepsilon_N(\eta)$ associated with a given $\eta$. Indeed, as $a \gtrsim 1$, Theorem 2 implies that $\varepsilon_N(\eta)$ scales with $N^{\frac{1}{a}} \lesssim N$ for $\varepsilon \geq c_3$.

## 4.2 Experiment 2: The Effect of the Wasserstein Ball

In the second simulation experiment we study the statistical properties of the out-of-sample logloss. As in [2], we set $n = 10$ and assume that the features follow a multivariate standard normal distribution, while the conditional distribution of the labels is of the form (1) with $\beta$ sampled uniformly from the unit sphere. We use the $\ell_2$-norm in the feature space, and we set $\kappa = 1$. All results reported here are averaged over 100 simulation runs. In each trial, we use $N = 10^2$ training samples to calibrate problem (6) and $10^4$ test samples to estimate the logloss distribution of the resulting classifier.

Figure 2(a) visualizes the conditional value-at-risk (CVaR) of the out-of-sample logloss distribution for various confidence levels and for different values of $\varepsilon$. The CVaR of the logloss at level $\alpha$ is defined as the conditional expectation of the logloss above its $(1 - \alpha)$-quantile, see [29]. In other words, the CVaR at level $\alpha$ quantifies the average of the $\alpha \times 100\%$ worst logloss realizations. As expected, using a distributionally robust approach renders the logistic regression problem more 'risk-averse', which results in uniformly lower CVaR values of the logloss, particularly for smaller confidence levels. Thus, increasing the radius of the Wasserstein ball reduces the right tail of the logloss distribution. Figure 2(c) confirms this observation by showing that the cumulative distribution function (CDF) of the logloss converges to a step function for large $\varepsilon$. Moreover, one can prove that the weight vector $\hat{\beta}$ tends to zero as $\varepsilon$ grows. Specifically, for $\varepsilon \geq 0.1$ we have $\hat{\beta} \approx 0$, in which case the logloss approximates the deterministic value $\log(2) = 0.69$. Zooming into the CVaR graph of Figure 2(a) at the end of the high confidence levels, we observe that the 100%-CVaR, which coincides in fact with the expected logloss, increases at *every* quantile level; see Figure 2(b).

## 4.3 Experiment 3: Real World Case Studies and Risk Estimation

Next, we validate the performance of the proposed distributionally robust logistic regression method on the MNIST dataset [30] and three popular datasets from the UCI repository: Ionosphere, Thoracic Surgery, and Breast Cancer [31]. In this experiment, we use the distance function of Definition 2 with the $\ell_1$-norm. We examine three different models: logistic regression (LR), regularized logistic regression (RLR), and distributionally robust logistic regression with $\kappa = 1$ (DRLR). All results reported here are averaged over 100 independent trials. In each trial related to a UCI dataset, we randomly select 60% of data to train the models and the rest to test the performance. Similarly, in each trial related to the MNIST dataset, we randomly select $10^3$ samples from the training dataset, and test the performance on the complete test dataset. The results in Table 1 (top) indicate that DRLR outperforms RLR in terms of CCR by about the same amount by which RLR outperforms classical LR (0.3%–1%), consistently across all experiments. We also evaluated the out-of-sample CVaR of logloss, which is a natural performance indicator for robust methods. Table 1 (bottom) shows that DRLR wins by a large margin (outperforming RLR by 4%–43%).

In the remainder we focus on the Ionosphere case study (the results of which are representative for the other case studies). Figures 3(a) and 3(b) depict the logloss and the CCR for different Wasserstein radii $\varepsilon$. DRLR ($\kappa = 1$) outperforms RLR ($\kappa = \infty$) consistently for all sufficiently small values of $\varepsilon$. This observation can be explained by the fact that DRLR accounts for uncertainty in the label, whereas RLR does not. Thus, there is a wider range of Wasserstein radii that result in

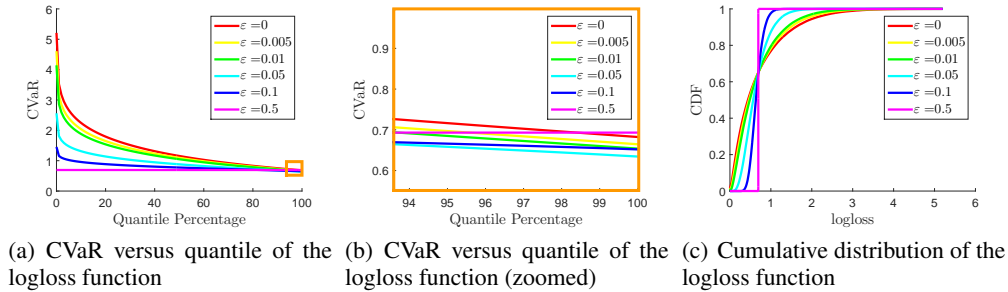

(a) CVaR versus quantile of the logloss function    (b) CVaR versus quantile of the logloss function (zoomed)    (c) Cumulative distribution of the logloss function

Figure 2: CVaR and CDF of the logloss function for different Wasserstein radii $\varepsilon$

Table 1: The average and standard deviation of CCR and CVaR evaluated on the test dataset.

|  |  | LR | RLR | DRLR |
|---|---|---|---|---|
| CCR | Ionosphere | $84.8 \pm 4.3\%$ | $86.1 \pm 3.1\%$ | $87.0 \pm 2.6\%$ |
|  | Thoracic Surgery | $82.7 \pm 2.0\%$ | $83.1 \pm 2.0\%$ | $83.8 \pm 2.0\%$ |
|  | Breast Cancer | $94.4 \pm 1.8\%$ | $95.5 \pm 1.2\%$ | $95.8 \pm 1.2\%$ |
|  | MNIST 1 vs 7 | $97.8 \pm 0.6\%$ | $98.0 \pm 0.3\%$ | $98.6 \pm 0.2\%$ |
|  | MNIST 4 vs 9 | $93.7 \pm 1.1\%$ | $94.6 \pm 0.5\%$ | $95.1 \pm 0.4\%$ |
|  | MNIST 5 vs 6 | $94.9 \pm 1.6\%$ | $95.7 \pm 0.5\%$ | $96.7 \pm 0.4\%$ |
| CVaR | Ionosphere | $10.5 \pm 6.9$ | $4.2 \pm 1.5$ | $3.5 \pm 2.0$ |
|  | Thoracic Surgery | $3.0 \pm 1.9$ | $2.3 \pm 0.3$ | $2.2 \pm 0.2$ |
|  | Breast Cancer | $20.3 \pm 15.1$ | $1.3 \pm 0.4$ | $0.9 \pm 0.2$ |
|  | MNIST 1 vs 7 | $3.9 \pm 2.8$ | $0.67 \pm 0.13$ | $0.38 \pm 0.06$ |
|  | MNIST 4 vs 9 | $8.7 \pm 6.5$ | $1.45 \pm 0.20$ | $1.09 \pm 0.08$ |
|  | MNIST 5 vs 6 | $14.1 \pm 9.5$ | $1.35 \pm 0.20$ | $0.84 \pm 0.08$ |

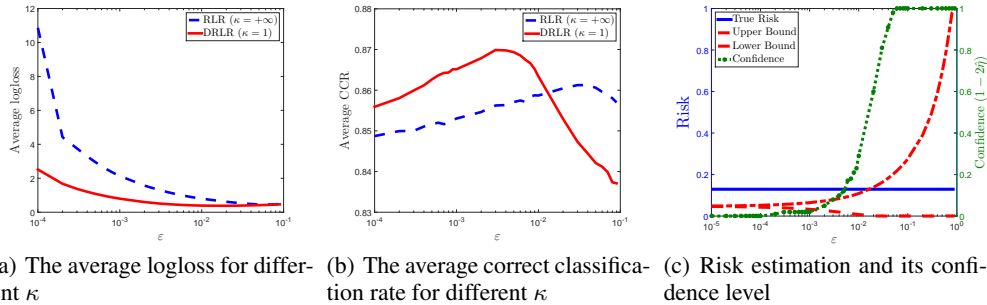

(a) The average logloss for different $\kappa$

(b) The average correct classification rate for different $\kappa$

(c) Risk estimation and its confidence level

Figure 3: Average logloss, CCR and risk for different Wasserstein radii $\varepsilon$ (Ionosphere dataset)

an attractive out-of-sample logloss and CCR. This effect facilitates the choice of $\varepsilon$ and could be a significant advantage in situations where it is difficult to determine $\varepsilon$ a priori.

In the experiment underlying Figure 3(c), we first fix $\hat{\beta}$ to the optimal solution of (7) for $\varepsilon = 0.003$ and $\kappa = 1$. Figure 3(c) shows the true risk $\mathfrak{R}(\hat{\beta})$ and its confidence bounds. As expected, for $\varepsilon = 0$ the upper and lower bounds coincide with the empirical risk on the training data, which is a lower bound for the true risk on the test data due to over-fitting effects. As $\varepsilon$ increases, the confidence interval between the bounds widens and eventually covers the true risk. For instance, at $\varepsilon \approx 0.05$ the confidence interval is given by $[0, 0.19]$ and contains the true risk with probability $1 - 2\hat{\eta} = 95\%$.

**Acknowledgments:** This research was supported by the Swiss National Science Foundation under grant BSCGI0_157733.

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
