[Reviews · NeurIPS 2015]

Submitted by Assigned_Reviewer_1

This paper uses the tools of distributionally robust optimization to analyse regularized logistic regression. In particular, the authors give an interpretation of l1-regularized regression as a robust optimization problem on the set of probability measures. They also introduce a new regularization which is proved to be also tractable.

This paper combines several recent mathematical techniques to fulfil this goal. The authors show strong skills in robust optimization. When the ambiguity set is defined with the Wasserstein metric, one obtains clear and powerful results. A technical difficulty of the paper is that when writing the distributionally robust problem, one initially gets either an infinite dimensional convex problem or a non-convex problem. Most of the 5-page appendix is dedicated to the reformulation of these untractable problems to finite dimensional convex problems which are thus tractable.

I have a few remarks for improvements. Page 2, lines 70-76: It looks like you are suggesting that the 2-norm is differentiable while it is not (the squared 2-norm is differentiable though) Appendix page 1, line 38: if 0 in [0,1] -> if theta in [0,1] Appendix page 2, line 87: only continuOUS components Appendix page 3, line 131: You should have kappa instead of kappa/N (you divided twice by N)

Summary: This paper uses the tools of distributionally robust optimization to analyse regularized logistic regression. They introduce a new regularization with a clear theoretical justification and algorithmic Framework.

Submitted by Assigned_Reviewer_2

This paper proposes a distributionally robust approach to logistic regression, which minimizes a worst-case expected logloss function among all distributions in the Wasserstein ball. It is an interesting research direction to give a robustness interpretation for ad hoc regularization techniques. However, I have still some concerns regarding the proposed model.

1: How can we interpret regularization techniques using distributionally robust perspective? Remark 1 discusses that the regularization coefficient coincides with the radius of the Wasserstein ball, but it holds only in the case of kappa=\infty.

2: I could not understand the motivation using the Wasserstein metric instead of other metrics (e.g., Bregman divergences, KL-divergence) in order to evaluate the difference between two probability distributions.

3: Section 3.3 discusses the best-case expectation as well as the worst-case expectation. Can the authors do numerical experiments for the best-case expectation model? It would be interesting to compare the average CCRs (test accuracies) of the worst- and best-case expectations over Wasserstein balls.

4: The proposed model includes two parameters, \kappa and \epsilon, and the flexibility in choosing the norm as described in Definition 2. How to choose parameter values and the norm? Among three experiments, different norms are used without specific reasons; L_\infty in Experiment 1, L2 in Experiment 2 and L1 in Experiment 3.

5: Additional numerical experiments using more UCI datasets may be necessary. Currently three UCI datasets are used, but three datasets are not enough to show the performance of the proposed model.
Summary: It is an interesting research direction to give a robustness interpretation for ad hoc regularization techniques. However, numerical experiments given in the paper are not so convincing.

Submitted by Assigned_Reviewer_3

This paper presents distributionally robust logistic regression (DRLR); the uncertainty set under consideration here is the Wasserstein ball of distributions, centered at the empirical (feature, label) distribution.

The authors reformulate the problem setup so that it leads to a tractable convex optimization problem.

The authors go on to provide a generalization error guarantee (which depends, counter-intuitively, on the assumption that the data-generating distribution is not heavy-tailed), and also present (tractable) convex optimization problems for computing upper and lower bounds on this quantity.

The authors present simulation results confirming the intuitive behavior of DRLR, and also show (on real data) that the method (essentially) doesn't hurt.

Overall, I thought this was a nice contribution to the growing line of work on distributional robustness, and is likely to spur follow-on work.

The paper was very well written (although there were a few typos scattered throughout, e.g., "week" --> "weak" on L62).

I did, however, have a few comments re: the empirical evaluation:

- Firstly, it is not really that surprising (to me, anyway) that your accuracy is essentially the same as that of your competitors on the real data (since typically, you have to sacrifice something when you gain in robustness) -- did you consider reporting CVaR on the test set for all the methods (you might see a better gain here, and this might more precisely characterize the performance of your method)?

- Secondly, could you provide some discussion on how tractable the optimization problems (7, 10a, 10b) actually are?

What are the specs of the computer you are using to solve these problems?

Do you have any timing results?

- How sensitive is your method to the choice of norm (you seem to use the Linfinity, L1, and L2 norm in your experiments, which indicates to me that no one norm is suitable for all problems) and kappa?

Is the guidance for selecting these hyper-parameters simply (some kind of) cross-validation (which, btw, adds significantly to the computational cost, but does hinge on my previous question re: timing results)?

- Part of your (stated) motivation for pursuing robustness is sparsity in the design matrices -- I wonder if you could conduct simulations where you play with fat vs. skinny design matrices with different levels of sparsity.

- I wonder if you could conduct some simulations to check the accuracy of Thm. 2, as the light-tailed assumption does seem restrictive to me (it is good that you included Fig. 3c in the paper).
Summary: Overall, I thought this was a nice contribution to the growing line of work on distributional robustness, and is likely to spur follow-on work.

Some of the experimental evaluation could be fleshed out and analyzed a bit more, but I thought this was ultimately a well-done paper.

Author Feedback
Author rebuttal: We thank the reviewers very much for their appreciation of our results and their constructive criticism, which helps us to improve the paper significantly.

Differences to the robust approach: A key advantage of our distributionally robust approach is that it offers a strong out-of-sample performance guarantee, which is facilitated by powerful results from statistics. It seems that similar guarantees would be more difficult to obtain with a robust approach. Moreover, we can prove that our approach is equivalent to perturbing the data points much like in the robust setting. However, the sum of probability-weighted absolute perturbations from the data points must be bounded by the Wasserstein radius. Thus, large perturbations are given less weight. In the robust approach all perturbations reside in identical balls and are thus treated alike.

Interpretation of regularized logistic regression (RLR): RLR is a special case of our approach if kappa is infinite. This means that data points with different labels are assigned an infinite distance. Thus, the Wasserstein distance between two distributions is infinite if they assign different labels to a fixed feature vector with positive probability. Any distribution in a Wasserstein ball around the empirical distribution must then have non-overlapping conditional supports for y=+1 and y=-1. Thus, setting kappa to infinity reflects the belief that the label is a (deterministic) function of the feature and that label measurements are exact. As this belief is not tenable in most applications, our approach (with finite kappa) may be more satisfying.

Choice of the probability metric: Good probability metrics offer strong out-of-sample performance guarantees, lead to tractable optimization problems, and enjoy asymptotic consistency properties (i.e., the radius of a confidence ball around the empirical distribution drops to zero as N increases). The Wasserstein metric scores high on all three criteria. In contrast, the Bregman or KL distance between any discrete and any continuous distribution is always infinite. Any bounded Bregman/KL-ball centered at the empirical distribution thus contains no continuous distribution. This is troubling as we expect physical distributions to be continuous, and it may prevent us from finding strong out-of-sample guarantees.

Best-case approach: We could minimize the best-case instead of the worst-case expected logloss. While conceptually attractive, this would lead, however, to an intractable nonconvex optimization problem. But we can efficiently compute the best-case and worst-case CCR for any fixed classifier as described in Sections 3.3 (theory) and 4.3 (experiment).

Choice of design parameters: The norm in the feature space and the parameter kappa determine the metric in the feature-label space. The choice of good metrics is a difficult problem in its own right (see e.g. Weinberger & Saul, JMLR, 2009 or Xing et al., NIPS, 2002). In the experiments we used different norms to showcase the flexibility of our scheme. We have now rerun all experiments with the l1-norm, which is attractive in terms of tractability. The results have not changed noticeably. The parameter kappa reflects our relative trust in the label measurements (see also the comment on RLR above). We emphasize that kappa=1 is pre-specified in all experiments; it is not selected by time consuming cross-validation. This choice systematically improved performance relative to RLR across all experiments. The choice of epsilon is guided by Theorem 2 and depends on the desired confidence eta and the sample size N (see also Figure 1).

Numerical results: As suggested by Reviewer 7, we tested our method on challenging MNIST datasets (see table below). The results show that our approach outperforms RLR in terms of CCR by about the same amount by which RLR outperforms classical LR (0.3%-1%), consistently across all experiments. Following a suggestion by Reviewer 3, we also evaluated the out-of-sample CVaR of logloss, which is a natural performance indicator for robust methods. As anticipated, our approach wins by a large margin (outperforming RLR by 4%-43%). All experiments were run on an Intel XEON CPU (3.40GHz). For the largest instance studied (N=1000), the problems (2), (3), (7) and (10) were solved in 2.1, 4.2, 9.2 and 0.05 seconds, respectively.

Light-tailed assumption: While the assumption may seem restrictive, we think that it is of little concern in practice. Most real data series have natural bounds determined by physics. We tried to illustrate Theorem 2 through the experiments in Section 4.1; see Figure 1.

In the revision we will clarify all of the above points and include all new numerical results. Thank you again for the valuable feedback.

Table: MNIST results
CCR | RL |RLR |DRLR
1 vs 7|97.8|98.0|98.6
4 vs 9|93.7|94.6|95.1
5 vs 6|94.9|95.7|96.7

CVaR | RL |RLR|DRLR
1 vs 7|3.9 |0.7|0.4
4 vs 9|8.7 |1.4|1.1
5 vs 6|14.1|1.3|0.8